# An instrument to assess HIV-related knowledge and adjustment to HIV+ status, and their association with anti-retroviral adherence

**M. Barton Laws****[1]\*, Yoojin Lee[1], William S. Rogers[2], Tatiana Taubin[1], Ira B. Wilson[1]**

**1** Dept., of Health Services, Policy and Practice, Brown University School of Public Health, Providence, RI, United States of America, **2** Institute for Clinical Research and Health Policy Studies, Tufts University, Boston, MA, United States of America

\* Michael_Barton_Laws@brown.edu

## Abstract

### Background

Findings on the association between health literacy and anti-retroviral (ARV) adherence are inconsistent. Health literacy is usually operationalized with simple tests of basic literacy, but more complex conceptions of health literacy include content knowledge. People living with chronic illness also conceptualize and experience illness in ways other than biomedical or mechanistic models of disease.

### Objective

There are no instruments that comprehensively assess knowledge of people living with HIV concerning HIV disease and treatment; or psychological adjustment to being HIV+. Little is known about the relationship between factual knowledge, or positive identification as HIV+, and anti-retroviral (ARV) adherence.

### Methods

Formative work with in-depth semi-structured interviews, and cognitive testing, to develop a structured instrument assessing HIV-related knowledge, and personal meanings of living with HIV. Pilot administration of the instrument to a convenience sample of 101 respondents.

### Key results

Respondents varied considerably in their expressed need for in-depth knowledge, the accuracy of their understanding of relevant scientific concepts and facts about ARV treatment, and psychological adjustment and acceptance of HIV+ status. Most knowledge domains were not significantly related to self-reported ARV adherence, but accurate knowledge specifically about ARV treatment was (r = 0.25, p = .02), as was an adapted version of the Need for Cognition scale (r = .256, p = .012). Negative feelings about living with HIV (r = .33, p =

**Data Availability Statement:** All survey response data and the original survey instrument are available at the Brown Digital Repository, https://doi.org/10.26300/pmqv-td23.

**Funding:** This work was funded by a grant from the National Institute of Mental Health, Grant # 5R21MH092781-02, to MBL. https://www.nimh. nih.gov/index.shtml The funders had no role in the study design, data collection and analysis, decision to publish, or preparation of the manuscript.

**Competing interests:** The authors have declared that no competing interests exist.

.0012), and medication taking (r = .276, p = .008) were significantly associated with non-adherence.

## Conclusion

The instrument may be useful in diagnosing addressable reasons for non-adherence, as a component of psychoeducational interventions, and for evaluation of such interventions.

## Introduction

Consistent adherence to antiretroviral (ARV) regimens by people living with HIV (PLWH) is of compelling importance to their own long-term health [1, 2], and to public health [3–6]. Among patient characteristics commonly hypothesized to be associated with ARV adherence is health literacy, which in practice is typically measured by simple tests of general reading ability. Some studies have concluded that low health literacy, operationalized using the REALM [7] that tests pronunciation of medical terms, or the TOFHLA [8], a test of reading comprehension, have found that low literacy is a significant independent predictor of non-adherence [9–12]. However, other studies have found that health literacy is not associated with ARV adherence [13], or indeed that lower health literacy is positively associated with ARV adherence [14]. Others have concluded that any association between health literacy and ARV adherence is due in whole or part to confounding factors such as self-efficacy [15] or stigma [16].

A review concluded that understanding of the connections between health literacy and outcomes was limited, and proposed a causal model incorporating numerous inputs and outputs to health literacy [17], which expands the definition of health literacy to include the *knowledge and beliefs* people hold about their health conditions. Information about the relationship of content knowledge to ARV adherence is scant, however. One observation is that some PLWH believe that ARVs should not be taken when a person drinks alcohol, which is associated with non-adherence [18, 19].

These inquiries are based implicitly on the assumption that people reason in a manner consistent with the Health Beliefs Model [20], balancing perceptions of the likelihood and costs of adverse events against the costs and benefits of adopting a protective behavior. However, people may make decisions based on other kinds of knowledge and reasoning. In medical sociology, it is conventional to distinguish between "disease" and "illness" [21–23]. Disease is a biomedical category, referring to entities which are similar regardless of the psychosocial setting or the afflicted individual. Illness refers to experience which is particular to the individual and patterned by psychological, social and cultural factors. In Kleinman's terms, both disease and illness incorporate explanatory models for processes and experiences [21, 23].

While patients' may explain pathophysiology, for example, in ways that do not accord with their physicians, the term nevertheless implies causation and mechanism. Other studies of illness experience use different framing. A commonly studied element of illness experience is stigma, defined by Goffman as "an attribute that makes [a person] different from others in the category of persons available for him to be, and of a less desirable kind" [24] (p. 3). Stigma has perhaps been the most common organizing frame for studies of the HIV illness experience [25].

Much of the literature on the experience of living with chronic illness other than HIV has been framed by identity theory, of which stigma is only one component [26, 27]. In one common formulation, the "self" is relatively stable, as people think of themselves as remaining the

same person throughout the life course; but "identity" is comprised of one's various social role relationships [28, 29]. Diagnosis of HIV can be highly disruptive of statuses and relationships, for example through forced disclosure of sexuality or illicit drug use to family members, and stigma.

There have been few studies of identity reformation in people with HIV since the availability of effective therapy in the late 1990s. One found that incorporation of the HIV diagnosis into identity included a period of what the author calls "denial" lasting as long as 5 years, after which respondents experienced a "turning point," accepted that they had HIV and moved on in the process [30]. This suggests that people who have yet to reach the turning point may not be motivated to engage in self-care including adhering to medication regimens.

In previous qualitative research, we found that most respondents had limited, largely erroneous understanding of the scientific concepts concerning HIV and HIV treatment. Nevertheless, accurate scientific understanding was not generally associated with treatment adherence, with the exception of some specific erroneous beliefs [31]. We also found that non-biomedical categories of meaning and explanation were more salient for most respondents and of far more importance in their treatment decision making and adherence. These could be understood in terms of identity formation [32]. In sum, motivation to adhere to HIV regimens may be associated with both accurate biomedical understanding; and ways in which taking anti-retroviral medications is experienced in a person's lifeworld.

The "need for cognition" construct in social psychology [33], defined as "a stable individual difference in people's tendency to engage in and enjoy effortful cognitive activity," [34] (p. 198) suggests that accurate factual knowledge would be more important in predicting self-care behavior for some people than for others.

Little research directly assesses how information, values and personal meanings shape the medication practice of people living with HIV. We therefore drew on our prior qualitative research to develop a structured assessment of the factual knowledge and beliefs of people living with HIV about HIV disease and treatment; the personal meanings of being HIV positive and HIV treatments, including negative and positive impacts on identity; and how these are associated with ARV adherence.

## Methods

### IRB approval

This study was approved by the Research Protections Office of Brown University, Protocol Entitled: Explanatory Models of Illness and Decision Heuristics in HIV Care (#1106000441). Written consent was obtained for in-person interviews. Responses over the Internet required electronic consent.

### Instrument development

Relevant formative work has been described elsewhere [31, 32]. Briefly, we conducted semi-structured interviews with a convenience sample of 32 people living with HIV in two cities in New England. The interview covered domains based on Kleinman's classic components of explanatory models. We harvested content from the interviews to create a structured instrument.

### Instrument content

We found that components of explanatory models could be essentially biomedical in character, though often inaccurate [31]; or could refer to other domains more pertinent to people's lived

experience such as responsibility or stigma [32]. As we analyzed the data, we classified responses in both categories, and created structured items to assess agreement with particular factual beliefs, or personal meanings. We organized the draft instrument as follows:

1. *Sociodemographics*: Demographic background and medical history.

2. *ARV adherence*: A previously validated scale based on an index of three items [35–37].

3. *Need for cognition*: As we found in the formative interviews that respondents expressed varying degrees of interest in technical understanding of HIV disease and treatment, and in making decisions based on formal reasoning, we included the short form of the need for cognition scale [38]. Chronbach's alpha for the NFC has been reported to be .89 or .90 depending on age group. Test-retest reliability has been reported to be r = .76. Higher levels of NfC are correlated with higher levels of cognitive abilities, estimated IQ, and education. [39]

4. *General feelings about HIV and ARV medications*: Items representing personal meanings of living with HIV and taking HIV medications, corresponding to elements of illness experience expressed in the qualitative interviews. (See Table 1 for retained items.) These were scored on a four-point scale from Strongly Agree to Strongly Disagree. Based on the formative research, we hypothesized that people who report more negative feelings about living with HIV would report poorer ARV adherence.

5. *Reasons for taking HIV medications*: Examples of the 6 items in this section are "I want to take my HIV medications because I've seen what happens to people who don't take them," and "I want to take my HIV medications because my doctor thinks they will help me." These are scored on a 3-point scale from Very Important to Not Important.

6. *Reasons why people might not want to take HIV medications*: Examples of the 10 items in this section are "I don't have to think about having HIV if I don't take the medications" and "It's good to give your body a break from the medications every once in a while." These were scored as the above section.

7. *HIV knowledge "quizzes"*: Four sets of items focusing on knowledge about HIV.

   a. *Factual statements about HIV*: A quiz on biomedical knowledge, and common conspiracy theories. Examples include "HIV infects specific cells in your blood that your body needs to fight infections," and "There is a cure for HIV, but they're keeping it a secret." These are scored on a 3-point scale from "Correct" to "Not at All Correct," with a "Don't Know" option. We assigned a truth value to each statement and scored answers as correct or not, with "don't know" counting as incorrect.

   b. *Knowledge about T-cell count and viral load*: We asked respondents to select their own most recent test results from a list of ranges. We then presented a series of factual statements about T-cell count and viral load, with the same response options as the previous section.

   c. *Statements about anti-retroviral medications*: Again, a knowledge quiz with a three point response option plus "Don't Know."

   d. *Knowledge about viral drug resistance*: We first asked if people had ever heard of the concept of "drug resistance." (In the formative interviews, many said they had not.) For those who had heard of it, we again presented a knowledge quiz with the same response options as the previous 4 sections.

**Table 1. Factor analysis of items pertaining to personal meanings of living with HIV.** he instrument, 1 represents strong agreement and 4 represents strong disagreement. Therefore negatively framed items have had scores reversed. A negative score on a positively framed item indicates agreement.(Original variable names are shown to reveal the location of deleted items.).

| | | Factor1 | Factor2 |
|---|---|---|---|
| Q2_1a_inv | Taking my HIV medications reminds me that I have the disease. | 0.460 | -0.082 |
| Q2_1c | In some ways, finding out that I had HIV was a good thing, because it made me make better choices. | 0.166 | -0.290 |
| Q2_1d_inv | Inverse: It makes me angry that I have to take medications. | 0.696 | 0.026 |
| Q2_1e | Learning I had HIV wasn't a death sentence, it was a new start on life. | 0.004 | 0.568 |
| Q2_1f_inv | Taking HIV medications reminds me that I made a mistake. | 0.681 | 0.208 |
| Q2_1g_inv | Taking my HIV medications makes me feel angry because somebody gave me the disease. | 0.661 | 0.153 |
| Q2_1h_inv | Having HIV makes me worry about my health. | 0.603 | 0.081 |
| Q2_1i_inv | Having HIV makes me worry about giving it to somebody else. | 0.458 | 0.328 |
| Q2_1j_inv | Having HIV makes me worry that people will reject me. | 0.551 | 0.194 |
| Q2_1k_inv | I'm scared I might die from AIDS. | 0.630 | 0.027 |
| Q2_1l_inv | I can't stand having to take so many medications. | 0.418 | 0.106 |
| Q2_1n_inv | HIV is likely to shorten my life. | 0.489 | 0.315 |
| Q2_1o | I can live a normal life with HIV. | 0.306 | 0.514 |
| Q2_1q | HIV is just a disease like any other, you can learn to cope with it. | 0.342 | 0.681 |
| Q2_1r | I don't mind if people know I have HIV, I'm very open about it. | 0.234 | 0.112 |
| Q2_1s_inv | I still haven't really accepted that I have HIV. | 0.410 | 0.170 |

Higher scores indicate disagreement with an item. Consequently Factor 1 can be interpreted as representing positive adjustment to living with HIV.

Correlation of scale score derived from Factor 1 with self-reported ARV adherence: r = -.33, p = .0012

Factor 2 is not significantly correlated with adherence.

## Cognitive testing

We recruited 6 additional respondents. The first author read each item to the respondents. After they responded, he asked them to reflect on why they chose their answer, whether there was any ambiguity or confusion about the meaning of the item, and if the response categories were adequate. The interviews were audio-recorded and transcribed for analysis. We made some minor modifications to question wording, instructions, and phrasing of responses as a result.

The most substantial modification was to the need for cognition scale. Respondents consistently said that their process for deciding on HIV treatment might be different from the way they make other decisions. Accordingly, we reworded items to refer specifically to HIV treatment. (See Table 4).

## Instrument pilot testing

We implemented the instrument as an on-line Computer Administered Survey, using Illume™ software (DatStat corporation). We had members of the Consumer Advisory Board at a local HIV clinic complete the instrument and provide comments. In response, we clarified some instructions. We then recruited respondents and implemented administration through a link placed on the website of a local AIDS Service Organization, allowing people to complete the instrument anonymously on-line; and by having a Research Assistant recruit respondents at

the clinic and administer the questionnaire as an interview. The only eligbility requirements were that people identified as HIV+ and were able to complete the questionnaire in English.

## Analyses

We computed descriptive statistics for all variables. For the knowledge questions, we computed the percentage of correct and incorrect answers for each section. For the scale pertaining to personal meanings or motivations, we conducted exploratory factor analyses using varimax rotation. We tested the correlation of scale and knowledge scores with self-reported adherence, and knowledge scores with need for cognition, using Pearson's r. We then derived a parsimonious instrument retaining those components that were significantly associated with self-reported adherence. We then tested possible causal pathways by testing the association of the 1 remaining knowledge scale with adherence while controlling for Need for Cognition, and the association between Need for Cognition and adherence while controlling for the knowledge score. Analyses were conducted using SAS 9.4, with significance set at $p < .05$.

## Results

### Participant characteristics

We administered the questionnaire in 2014. One hundred five people responded. Due to attrition among respondents who self-administered, 99 people completed section 4 and 94 completed section 6. Only 63 of the remaining respondents indicated that they had ever heard of the concept of drug resistance, of whom 61 completed section 7.

Of 105 total respondents, 45 (42.9%) indicated that they were diagnosed with HIV in 1995 or earlier; 18 more were diagnosed in 2000 or earlier. Four were newly diagnosed in 2014, with the others distributed throughout the intervening years. Only 1 respondent indicated not having a current ARV prescription. 71% of respondents were male, the mean age was 49 years with a range from 18 to 73. Fifty-three percent identified as non-Hispanic white, 19% as African American, 11.4% as Latino, 4.8% (5 individuals) as Cape Verdean, and 9 people as various other ethnicities. Nineteen individuals (18%) were foreign born.

### Item and scale characteristics

Responses to most items in the section *General feelings about HIV and medications*, had good variation, although one item, "Taking my HIV medications makes me grateful that I have access to them," elicited disagreement from only 3 respondents. We deleted it as uninformative. As expected, factor analysis revealed a first factor on which 12 statements representing successful adjustment to living with HIV and taking medications had the highest loads. (Because the response categories ranged from 1 for strong agreement to 4 for strong disagreement, and disagreement with negatively framed items was considered to represent positive adjustment, we Most of these loadings ranged from .489 to .69. We retained items with smaller loadings because they contributed to the association of the scale with self-reported ARV nonadherence. We originally included both negatively and positively framed versions of an item intended to assess internalized stigma: "I don't want most people to know that I have HIV," and "I don't mind if people know I have HIV, I'm very open about it." Both loaded fairly weakly on Factor 1, with opposite valence. We retained the second, positively framed item (with reversed scoring), because the derived scale that included it was more strongly correlated with self-reported adherence. Four items expressing positive adjustment to living with HIV loaded on a second factor. The item "Most of the time I try not to think about having HIV" was not associated with either factor. We deleted it. (See Table 1 for the retained items and

their factor loadings.) The composite score for Factor 1 (sum of responses scored from 1–4) was significantly associated with self-reported ARV adherence, r = -.33, p = .0012. (Higher scores were associated with non-adherence.) Although the four items representing positive adjustment were not correlated with adherence, we retained them to avoid encouraging a positive response set.

Responses to the section *"The importance of reasons for taking HIV medications"* were less variable, with most respondents endorsing all of the listed reasons. Since this section was uninformative, we deleted it.

There was more variation in the section "Possible reasons for not wanting to take HIV medications." (See Table 2). This constituted a scale with high internal consistency, which was significantly correlated with non-adherence, r = .276, p = .008.

For the *HIV knowledge quiz*, of the 10 items we deemed to have a definitely correct answer, the mean number of correct answers was 5.8. As we expected from formative research, accurate knowledge in this domain was not associated with ARV adherence, so we deleted the section. Respondents' knowledge about T-cell count and viral load was often poor. Again, however this knowledge was not associated with self-reported ARV adherence.

The mean score on the knowledge quiz about ARV treatment was 7.5 out of 12 items, with considerable variance (sd = 2.69). The score of correct answers on this section was significantly correlated with self-reported ARV adherence (r = .25, p = .021). (See Table 3).

More than one quarter of respondents said they had not heard of the concept of drug resistance. Of the 65 who went on to answer questions in this section, misconceptions were very common. There was no association between accurate knowledge about drug resistance and adherence. (All deleted items are available from the corresponding author.)

## Need for cognition

The score on the modified Need for Cognition scale was highly significantly correlated with the quiz on HIV treatment (r = .39, p = .0002). The Need for Cognition scale was also independently associated with ARV adherence (r = .256, p = .012). Controlling for the Need for

**Table 2. Reasons for not wanting to take ARVs and scale consistency.**

| Cronbach Coefficient Alpha with Deleted Variable | | |
|---|---|---|
| Variable | Correlation with total | Alpha |
| I don't want to take my HIV medications when I'm depressed. | 0.439 | 0.837 |
| If I can't get something to eat, I don't take my HIV medications. | 0.384 | 0.841 |
| I'm afraid somebody might see me taking my HIV medications. | 0.342 | 0.844 |
| There are some kinds of HIV medications I don't want to take because I know somebody who tried it and it didn't work for them, or it made them sick. | 0.507 | 0.832 |
| I don't want to take HIV medications because of side effects. | 0.646 | 0.821 |
| I don't think I should take HIV medications when I drink alcohol. | 0.588 | 0.826 |
| I don't think I should take HIV medications when I use hard drugs such as cocaine, heroin or opioids, or methamphetamine. | 0.615 | 0.824 |
| I don't have to think about having HIV if I don't take the medications. | 0.628 | 0.823 |
| I sometimes forget to take them. | 0.294 | 0.847 |
| I can't stand having to take so many medications. | 0.610 | 0.824 |
| I feel fine so I don't think I need to take any medications. | 0.539 | 0.829 |
| It's good to give your body a break from the medications every once in a while. | 0.525 | 0.830 |

Overall alpha .84. Correlation with adherence r = .276, p = .008

**Table 3. ARV knowledge quiz.**

| Item | "correct" answer |
|---|---|
| You shouldn't take ARVs when you drink alcohol | F |
| You shouldn't take ARVs if you don't have something to eat | F |
| You shouldn't take ARVs when you use cocaine or heroin and other opiates | F |
| If you take ARVs for a long time, you won't have to take them any more. | F |
| It's important to take your ARVs on schedule to keep a consistent level of the medications in your blood | T |
| If you keep taking ARVs for a long time, your body becomes saturated with the medication, and you should stop for a while | F |
| Different ARVs work in different ways, so a combination works best | T |

Mean % correct 58%

Correlation of % correct with self-reported adherence r = .25, p = .021

Cognition score, the correlation between the knowledge quiz and adherence was no longer significant (r = .17, p = .118). Controlling for the quiz score, the correlation between Need for Cognition and adherence was also attenuated (r = .196, p = .07). (See Table 4 for the adapted Need for Cognition scale.) The causal relationship among these variables is unclear, but it appears that the correlation of adherence with Need for Cognition is partly mediated by knowledge and partly independent.

## Discussion

This instrument identifies gaps in knowledge and misconceptions about HIV and HIV treatment; and measures more and less successful psychological adaptation to the HIV diagnosis and treatment burden. In the pilot sample, it appears that misconceptions about HIV treatment, and failure to successfully integrate living with HIV into a positive self-identity, are both associated with non-adherence to ARV regimens. Accurate knowledge about HIV pathophysiology and mechanisms of treatment, including understanding of viral drug resistance, were

**Table 4. Adapted need for cognition scale.**

| | Scoring direction |
|---|---|
| a. It is important to me to understand as much as I can about my diagnosis. | + |
| b. It is not important to me to understand how medications work; I do what the doctor tells me to do. | - |
| c. I invest a lot of time learning about HIV and HIV treatments. | + |
| d. I try to learn all the facts about medications and make up my own mind about taking them. | + |
| e. I don't like to think very much about my health and medical care. | - |
| f. I would rather do something that requires little thought than something that is sure to challenge my thinking abilities. | - |
| g. I like to understand the issues about my medical care and make my own decisions. | + |
| h. It's enough for me that a medication gets the job done; I don't care how or why it works. | - |
| i. Learning about health and diseases doesn't interest me very much. | - |
| j. I like to understand concepts and ideas about medical treatments. | + |

Correlation with self-reported ARV adherence r = .211, p = .039

not associated with non-adherence. The findings are consistent with hypotheses that emerged from our formative work.

The instrument may be useful to directly inform clinicians about patients' misconceptions that may undermine adherence, and psychological barriers to adherence. It may also be a useful tool for individual or group psycho-educational interventions for people living with HIV, as a way of provoking and structuring discussion, and as a pre-test/post-test evaluation.

For clinicians, the implication of this work is that it is important to correctly diagnose the reasons why people may be non-adherent to ARVs or fail to engage in treatment. Clinicians often respond to non-adherence by re-emphasizing the biomedical facts about HIV and treatment [40], but this is unlikely to be effective for many people.

Developing and evaluating interventions to promote adherence and other effective self-management is more likely to yield results if they can be tailored to individual needs. People who do not accept their HIV status and are not motivated to protect their health will not benefit from supports designed for people who need information, reminders or other practical aids to adherence. Conversely, people who are motivated to protect their health but hold misconceptions that cause them to behave discordantly with their physicians' beliefs about treatment may benefit from an educational or information intervention. This structured assessment is intended to make those diagnoses.

This study is limited in that it was conducted in a single city in the northeastern U.S., and most respondents attended a single clinic. However, it is the only HIV specialty clinic in the area so they are likely to be representative of at least the local population. As with any novel instrument, it will have to be tested in various settings and the associations we found will have to be confirmed.

## Acknowledgments

Our thanks to Laura Kogelman, M.D., and Aadia Rana, M.D.

## Author Contributions

**Conceptualization:** M. Barton Laws, Ira B. Wilson.

**Data curation:** Yoojin Lee, Tatiana Taubin.

**Formal analysis:** M. Barton Laws, Yoojin Lee, William S. Rogers.

**Funding acquisition:** M. Barton Laws.

**Investigation:** M. Barton Laws, Tatiana Taubin.

**Methodology:** M. Barton Laws, Ira B. Wilson.

**Project administration:** M. Barton Laws, Tatiana Taubin.

**Supervision:** M. Barton Laws.

**Writing – original draft:** M. Barton Laws.

**Writing – review & editing:** Tatiana Taubin, Ira B. Wilson.

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
