## [Decision Letter · Decision Letter 0]

31 Mar 2020

PONE-D-19-35554

An instrument to assess HIV-related knowledge and adjustment to HIV+ status, and their association with anti-retroviral adherence

PLOS ONE

Dear Dr Barton Laws,

Thank you for submitting your manuscript to PLOS ONE. After careful consideration, we feel that it has merit but does not fully meet PLOS ONE’s publication criteria as it currently stands. Therefore, we invite you to submit a revised version of the manuscript that addresses the points raised during the review process.

We would appreciate receiving your revised manuscript by 15th May 2020. To enhance the reproducibility of your results, we recommend that if applicable you deposit your laboratory protocols in protocols.io, where a protocol can be assigned its own identifier (DOI) such that it can be cited independently in the future. For instructions see: http://journals.plos.org/plosone/s/submission-guidelines#loc-laboratory-protocols

We look forward to receiving your revised manuscript.

Kind regards,

Kwasi Torpey, MD PhD MPH

Academic Editor

PLOS ONE

Journal Requirements:

https://link.springer.com/article/10.1007%2Fs10461-016-1358-1

In your revision ensure you cite all your sources (including your own works), and quote or rephrase any duplicated text outside the methods section. Further consideration is dependent on these concerns being addressed.

3. Please clarify in your Methods section whether the questionnaires are published under a CC-BY license, or whether you obtained permission from the publisher to reproduce the questionnaire in this manuscript. Please explain any copyright or restrictions on this questionnaire.

5. Your ethics statement must appear in the Methods section of your manuscript. If your ethics statement is written in any section besides the Methods, please move it to the Methods section and delete it from any other section. Please also ensure that your ethics statement is included in your manuscript, as the ethics section of your online submission will not be published alongside your manuscript.

Reviewers' comments:

Reviewer's Responses to Questions

**Comments to the Author**

1. Is the manuscript technically sound, and do the data support the conclusions?

Reviewer #1: Partly

Reviewer #2: Partly

2. Has the statistical analysis been performed appropriately and rigorously? 

Reviewer #1: Yes

Reviewer #2: Yes

3. Have the authors made all data underlying the findings in their manuscript fully available?

Reviewer #1: Yes

Reviewer #2: Yes

4. Is the manuscript presented in an intelligible fashion and written in standard English?

Reviewer #1: Yes

Reviewer #2: Yes

5. Review Comments to the Author

Reviewer #1: This manuscript addresses an important and timely topic. ARV adherence continues to be an issue in the management of patients with HIV. Identifying factors affecting adherence that can be modified is necessary. Thus, developing assessment tools is important particularly if they are useful to providers as well as to researchers. Concerns that this reviewer has with this paper include the lengthy introduction. It seems as if the material on health literacy is not necessary or could be summarized in a few sentences. A more direct case needs to be made for personal meaning, values, and information and their relation to adherence which is stated in the purpose for the pilot study/tool development. Given that this study is a pilot test of a new tool, the purpose needs to include assessing psychometric properties and the hypotheses that are stated in the findings. The section labeled “cognitive testing” is confusing. Possibly renaming it would add clarity. Possibly writing the paper in a more straightforward manner such as what was done first (items were identified under specific headings), then given to a group of six similar to those who would complete the tool which was followed by individuals from a Consumer Advisory Board. Once the items were refined, then participants were recruited to pilot the instrument. The findings are described and hypotheses tested with moderate correlations found to be significant. Including the alpha level of .05 in the analysis section would have been helpful. The discussion section needs to be tied back to the literature. One possible reason for why the hypotheses were significant may be that adherence was measured with a self-report tool and this new instrument is also self-report. Limitations of this pilot study and next steps/future directions need to be added. Given that only 105 persons were in the study and all did not complete all sections of the instrument, the implications for practice may need to be toned down until another study examining the tool is conducted.

Reviewer #2: The relationship between literacy and HIV adherence is well known. A strength of this study is that it incorporates facets of meaning, identity, and biomedical knowledge in creating a comprehensive assessment of knowledge among adults living with HIV. The paper has several other strengths, including providing detailed information about the pilot testing, incorporating the feedback of an advisory board, and their use of cognitive interviewing to improve the scale.

My main areas to improve the paper relates to the methodology and analysis.

General comments:

Referring to people living with HIV: Authors used a variety of labels HIV+; people with HIV, living with HIV. Since "people living with HIV" is the more acceptable term, they can consider using it consistently.

Methods

• The methods indicate that two modalities were used for data collection: online and through a research assistant. Comments are:

a. Were there any differences in the results by these two methods?

b. What is the interclass correlation (ICC) between those two modalities?

c. What proportion completed the survey online and in-person?

• What was the inclusion and exclusion criteria for the study?

Analysis

• Need for cognition scale: Since this scale is central to the results of the paper, the authors are encouraged to report relevant reliability and validity literature of the scale in the text. For example, what is the internal consistency and validity of the scale?

• In addition, they should also give some examples of items in the “Need for Cognition” scale, explain how it is scored, and interpreted.

• Their health literacy scale seems to have five sections (as described in the methods). It is unclear whether the factor analysis was done across all of the items. For example, only two factors were reported from the Factor Analysis, and the items do not include the knowledge quizzes. So it is unclear if their health literacy scale assessed only the personal meaning of living with HIV or also included the knowledge quizzes? If this is the case, why didn't they include the knowledge items, since they wanted to create a comprehensive assessment of health literacy?

• Relatedly, in Table 1, it may be helpful to have sub-categories, so that readers are aware of what items relate to what sections/domains of the scale.

• The analysis is very scant, and more details are needed. First, what was the analytical process for the factor analysis? What factor rotations were used? What indices of fitness were used to arrive at the two factors?

• In addition, in the results section, under the "Need for Cognition," they share the results of two separate analyses (page 11, lines 7 and 8). In one analysis, they controlled for quiz score, and another, for cognition. These analyses and the rationale behind them are not detailed in the methods section.

• The authors argue that this scale is more comprehensive. They encourage its use within clinical settings. What is missing, however, is the validity of the scale. Language about validity is missing. For example, I can see the relationship between literacy and ARV adherence as construct validity. They can also report on the discriminant validity of the scale, using items they know are not related to literacy.

• They should indicate the level of significance used and the statistical package for analysis.

• There is no justification for the sample size; thus, it is unclear whether the lack of association, for example, between biomedical knowledge and ARV adherence was due to a small sample size.

Results

• The authors did not provide any data on the socio-demographic and medical history of the participants. Since participant backgrounds may influence their knowledge and self-identity, it will be helpful here for them to report these results. For example, what is the level of education of the participants? How long have they been living with HIV? What are the age categories? What was the level of internalized stigma within the population (was any data collected here) since stigma is also an essential part of patient experience and identity

• The authors do not report the overall Cronbach's alpha of the scale, nor the sub-scales, although the individual alphas are reported.

• One page 9, line 17, they reported the mean of the ARV treatment items and commented that there was a considerable variance. They should report the variance.

Discussion

• The authors do not report any limitations of the study.

• Can they compare the reliability of their literacy measure to that of other studies?

• Comments on validity?

6. PLOS authors have the option to publish the peer review history of their article (what does this mean?). If published, this will include your full peer review and any attached files.

Reviewer #1: No

Reviewer #2: No

---

## [Author Response · Author response to Decision Letter 0]

31 May 2020

Response to Reviewers

Reviewers’ comments are in italics.

Reviewer 1

The relationship between literacy and HIV adherence is well known. The strength of this study is that it incorporates facets of meaning, identity, and biomedical knowledge in creating a comprehensive assessment of knowledge among adults living with HIV. The paper has several strengths, including providing detailed information about the pilot testing and incorporating the feedback of an advisory board. They also conducted cognitive interviewing to improve the scale.

Thank you. We hope we have shed more light on the relationship between literacy and HIV adherence. 

My main areas to improve the paper relates to the methodology and analysis.

General comments

Referring to people living with HIV: Authors used a variety of labels HIV+; people with HIV, living with HIV. Since "people living with HIV" is the more acceptable term, they can consider using it consistently.

We have consistently referred to PLWH after first spelling it out.

Methods

• The methods indicate that two modalities were used for data collection: online and through a research assistant. Comments are 

a. Were there any differences in the results by the two methods

b. What is the interclass correlation (ICC) between those two modalities?

c. What proportion completed the survey online and in-person

Only 20 people completed the questionnaire on-line. Given that small number, it is probably not informative to check the differences.

What was the inclusion and exclusion criteria for the study?

The only requirements were that people identified themselves as HIV positive and were able to complete the questionnaire in English. We have added this statement to the methods (line 8).

Analysis

• Need for cognition scale: Since this scale is central to the results of the paper, the authors are encouraged to report relevant reliability and validity literature of the scale in the text. For example, what is the internal consistency and validity of the scale?

Chronbach’s alpha for the NFC has been reported to be .89 or .90 depending on age group. Test-retest reliability has been reported to be r=.76. Higher levels of NfC are correlated with higher levels of cognitive abilities, estimated IQ, and education. (Soubelet and Salthouse, 2017). We have added this information. (page 5)

• In addition, they should also give some examples of items in the Need for cognition scale, explain how it is scored, and interpreted.

The scale is provided as Table 4, with the direction of scoring indicated.

• Their health literacy scale seems to have five sections (as described in the methods). It is unclear whether when the factor analysis was done was all of the items across the multiple domains included. For example, only two factors were reported from the Factor Analysis, and the items do not include the knowledge quizzes. So it is unclear if their health literacy scale assessed only the personal meaning of living with HIV or included the knowledge quizzes? If this is the case, why didn't they include the knowledge items, since they wanted to create a comprehensive assessment of health literacy?

As we explain in the results, three of the sections were not correlated with ARV adherence so we deleted them in order to create a parsimonious instrument. We did not factor analyze the knowledge quizzes because the answers are either true or false. The tables contain all of the items that we retained and show where the factor analysis was done. We realize this was not clearly explained in the methods, we have revised accordingly. 

• Relatedly, in Table 1, it may be helpful to have sub-categories, so that readers are aware of what items relate to what sections/domains of the scale.

We aren’t sure we understand this comment. We factor analyzed this scale and found a two-factor solution, which we show.

• The analysis is very scant, and more details are needed. First, what was the analytical process for the factor analysis? What rotations did they use? What indices of fitness did they use to arrive at the two factors? 

We used varimax orthogonal rotation, which simplifies the interpretation of the factors by maximizing the variances of the squared loadings for each factor. The interpretation of the two factors is intuitively obvious, and the first factor is associated with ARV adherence. 

• In addition, in the results section, under the "need for cognition," they share the results of two separate analyses (page 11, lines 7 and 8). In one analysis, they controlled for quiz score, and another, for cognition. These analyses and the rationale behind them are not detailed in the methods section. 

Again, we have revised the methods to explain this more clearly. (Page 7)

• The authors argue that this scale is more comprehensive. They encourage its use within clinical settings. What is missing, however, is the validity of the scale. Language about validity is missing. For example, I can see the relationship between literacy and ARV adherence as construct validity. They can also report on the discriminant validity of the scale, using items they know are not related to literacy. 

There are actually four scales in our instrument, as shown in the four tables. Each of them is associated with self-reported adherence, which constitutes construct validity. We believe that it is responses to individual items, particularly on the knowledge quizzes, that is most likely predictive, rather than the overall score, i.e. a particular misconception underlies non-adherence. The usefulness in a clinical setting is then to diagnose the reasons for non-adherence in particular individuals, which may be addressable. 

• They should indicate the level of significance used and the statistical package for analysis. 

0.05, SAS 9.4 We have added this information to the methods section.

• There is no justification for the sample size; thus, it is unclear whether the lack of association, for example, between biomedical knowledge and ARV adherence was due to a small sample size. 

Given that we had no basis for assessing prior probability, we could not do a power analysis in advance. The sample size was the largest we could obtain within our budget, and it is typical of pilot tests of instruments. There was not even a trend for association between the items we deleted and adherence. If there is an association it is very small. This observation is consistent with our formative qualitative research, as referenced. (Laws, et al, 2015)

Results

• The authors did not provide any data on the socio-demographic and medical history of the participants. Since participant backgrounds may influence their knowledge and self-identity, it will be helpful here for them to report these results. For example, what is the level of education of the participants? How long have they been living with HIV? What are the age categories? What was the level of internalized stigma within the population (was any data collected here) since stigma is also an essential part of patient experience and identity

We provide this information on page 8 under “Participant characteristics, with the exception of internalized stigma which we did not assess as such. However, items Q2-1j and Q2_1r in table 1 are direct references to stigma.

• The authors do not report the overall Cronbach's alpha of the scale, nor the sub-scales, although the individual alphas are reported. 

Overall alpha of the scale shown in table 2 is .84. We have added this information.

 

• One page 9, line 17, they reported the mean of the ARV treatment items and commented that there was a considerable variance. They should report the variance. 

The sd was 2.69. We have added this information.

Discussion

• The authors do not report any limitations of the study.

We have added a statement about limitations.

• Can they compare the reliability of their literacy measure to that of other studies? 

We are unaware of the existence of any measures of HIV-related knowledge (other than questionnaires about transmissibility) for which reliability has been assessed. 

• Comments on validity?

We do show the association of our measures with self-reported adherence, which speaks to validity.

---

## [Editor Report · Decision Letter 1]

2 Jun 2020

An instrument to assess HIV-related knowledge and adjustment to HIV+ status, and their association with anti-retroviral adherence

PONE-D-19-35554R1

Dear Dr. Barton Laws,

We’re pleased to inform you that your manuscript has been judged scientifically suitable for publication and will be formally accepted for publication once it meets all outstanding technical requirements.

Kind regards,

Professor Kwasi Torpey, MD PhD MPH

Academic Editor

PLOS ONE

Additional Editor Comments (optional):

Comments satisfactorily addressed
---

## [Editor Report · Acceptance letter]

11 Jun 2020

PONE-D-19-35554R1 

An instrument to assess HIV-related knowledge and adjustment to HIV+ status, and their association with anti-retroviral adherence 

Dear Dr. Laws:

I'm pleased to inform you that your manuscript has been deemed suitable for publication in PLOS ONE. Congratulations! Your manuscript is now with our production department. 

Kind regards, 

on behalf of

Professor Kwasi Torpey 

Academic Editor

PLOS ONE